

# A new device to mount portable energy dispersive X-ray fluorescence spectrometers (p-ED-XRF) for semi-continuous analyses of split (sediment) cores and solid samples

Philipp Hoelzmann[1], Torsten Klein[1], Frank Kutz[1], Brigitta Schütt[1]

[1]Physical Geography, Institute of Geographical Sciences, Department of Earth Sciences, Freie Universität Berlin, Berlin, Germany

*Correspondence to*: Philipp Hoelzmann (philipp.hoelzmann@fu-berlin.de)

**Abstract.** Portable energy-dispersive X-ray fluorescence spectrometers (p-ED-XRF) have become increasingly
popular in sedimentary laboratories to quantify the chemical composition of a range of materials such as sediments, soils, solid samples, and artefacts. Here, we introduce a low-cost, clearly arranged unit that functions as a sample chamber (German industrial property right no. 20 2014 106 048.0) for p-ED-XRF devices to facilitate economic, non-destructive, fast, and semi-continuous analysis of (sediment) cores or other solid samples. The spatial resolution of the measurements is limited to the specifications of the applied p-ED-XRF device – in our case a
Thermo Scientific NITON XL3t p-ED-XRF spectrometer with a maximum spatial resolution of 0.3 cm and equipped with a charge-coupled device (CCD)-camera to document the measurement spot. We demonstrate the strength of combining p-ED-XRF analyses with this new sample chamber to identify Holocene facies changes (e.g. marine vs terrestrial sedimentary facies) using a sediment core from an estuarine environment in the context of a geoarchaeological investigation at the Mediterranean coast of southern Spain.

## 1 Introduction


Bulk-sediment chemistry data of split core sediment surfaces provided by non-destructively measured portable X-ray fluorescence is important for the determination of sedimentary composition, the interpretation of varying facies, and thus for the reconstruction of environmental conditions. Therefore, computer-controlled core scanning XRF tools are often successfully applied (Jansen et al. 1998; Koshikawa et al. 2003; Kido et al. 2006; Richter et
al. 2006) and supply bulk-sedimentary data at nearly continuous μm-scale resolution, first of all for environments that provide highly resolved archives with low sediment accumulation rates such as in cores from marine or lacustrine environments. However, these instruments are extremely cost-intensive and especially terrestrial sediments often show a lower temporal resolution with much higher sediment accumulation rates in the range of centimetres per year that allow the application of basic XRF-instruments with much lower spatial resolution.
This paper introduces a hands-on, low-cost device (German industrial property right no. 20 2014 106 048.0) that uses common adapters to mount p-ED-XRF devices so that these can provide bulk-sedimentary chemistry data from non-destructive measurements at the surface of a split sediment core or from other solid samples at a spatial



resolution between 0.3 cm * 0.3 cm to 0.8 cm * 0.8 cm. Our device bridges the gap between sophisticated, computer-controlled core scanning XRF systems with highest spatial resolution and simple manually performed hand-held p-ED-XRF analyses. We demonstrate the strength of combining p-ED-XRF analyses with this new sample chamber to identify Holocene facies changes (e.g. marine vs terrestrial sedimentary facies) using a

sediment core from an estuarine environment in the context of a geoarchaeological investigation at the Mediterranean coast of southwestern Spain.

At the archaeological site Ayamonte (S-Spain), located at the estuary where the Guadiana River flows into the Atlantic in the southwest of the Iberian Peninsula, sediment archives were investigated in order to provide input for an improved environmental reconstruction of the Holocene settling phases. In our study the bulk-sedimentary

chemistry data obtained in cm-resolution by p-ED-XRF was used as an integral measure to differentiate estuarine/lagoonal from terrestrial facies in a series of sediment drillings of up to 900 cm depth. We present quantitative element data at ca. 1 cm resolution from one of the analyzed sediment cores to point out the low-cost, fast, non-destructive, and accurate results obtained with the mounted p-ED-XRF spectrometer.

## 2 Study Site, Material, and Methods

### 2.1 Study Site

The new discovery of the Phoenician necropolis and colony Ayamonte opened a new chapter in Phoenician research at the southwestern coasts of the Iberian Peninsula (Teyssandier and Marzoli, 2014). To contribute towards an understanding of the utilization of the ancient settlement, initial studies focus on the feasibility of a rearward harbour location within the lagoonal-like environments of the "Estuario de la Nao". Sediment-drillings

were performed at suspected sites and the sediments were dated and analyzed for their sedimentological and geochemical composition.

The study site is located in the Spanish province of Huelva at the Atlantic coast in the southwest of the Iberian Peninsula and comprises the small tributary "Estuario de la Nao", which discharges into the lower Guadiana estuary, located approx. 3 km north of the city of Ayamonte. AYA05 (37°14'4.61''N, 7°23'45.38''W) is a core of

a series of three drilling sites (Fig. S1) and is located at the footslope of a hoof-shaped hillside within a salt marsh area, which is inundated during spring high tides (Fletcher, 2005). The "Estuario de la Nao" drains an area of approx. 10 km² and its floodplain contains agricultural, industrial, and semi-natural environments. A detailed report with the geoarchaeological results has been published in Klein et al. (2016 in press). Here, we focus on the analyses that were performed with the newly designed sample chamber (German industrial property right no. 20

2014 106 048.0) with the mounted p-ED-XRF spectrometer to support sedimentary facies differentiation.

### 2.2 Material

The sediment core AYA05 has a total depth of 900 cm and consists of organic-rich, very carbonate-poor gray (7Y 5/1) to grayish-brown (8,5YR 5/2) pure silt (Fig. S2). Distinct layers of marine fauna, macro-botanical remains and charred wood occur as well as four layers with broken shell material (at 7.20 to 7.12 m below surface (m b.s.);

6.85 to 6.75 m b.s.; 6.35 to 5.80 m b.s.; 5.65 to 5.30 m b.s.). The age-model for core AYA05 consists of five stratigraphic radiocarbon dates from charred wood or plant remains that show no age-inversions and cover the





period from c. 5000 to 3000 cal a BP. For a detailed description of all cores and corresponding age-models refer to Klein et al. (2016, in press).

For the p-ED-XRF measurements no laborious core preparations were necessary. The 1 m long core sections were split into two halves: an air-dried half was covered with a mylar foil (0.4 μm thick), transferred to the ASC® manual

core track with the mounted sample chamber and after the p-ED-XRF spectrometer was fixed to the sample chamber the core was ready for analysis.

### 2.3 Methods

The sample chamber (German industrial property right no. 20 2014 106 048.0) that holds the measurement system (e.g. p-ED-XRF spectrometer; Fig. 1a) for the analysis of sediment and drill core samples comprises a housing-

base and housing-cover. Both base and cover are equipped with cladding and are constructed so that they form a plugin-device (Anonymous, 2014) that consists of radiation shielding material (metal alloy). The cladding of the housing-base and the housing-cover has an opening with a lead-through for the holding fixture of the split sediment or drill core. In order to analyze the samples, there is an aperture in the bottom of the housing cover for the arrangement of the measurement system. The analyzer or measurement system is placed above the sample so that,

in the case of XRF spectrometers, the X-ray source is directed from the top down on the split sediment or drill core or solid sample (Fig. 1b). Cores of undefined length or solid samples can be passed through the sample chamber on the holding track.

The ASC® manual core track (http://www.ascscientific.com/mantrack.html) accommodates the dried and split sediment cores so these slide in a low friction track and are positioned at each measurement point by the operator.

The sample chamber is adjusted to the core track in such a way that quasi-continuous measurements at a spatial resolution down to 0.3 cm * 0.3 cm become possible for different types of samples, cores or "u-channels" (Fig. 2). Our measurements were performed using a Thermo Scientific NITON XL3t p-ED-XRF spectrometer equipped with a charge-coupled device (CCD) camera to document the measurement spot and a semi-conductor detector for the detection of the XRF. A spatial resolution of 0.8 cm * 0.8 cm was selected for the analyses. Following

measurement conditions were applied:

| | | |
|---|---|---|
| X-Ray source: | Ag-anode | |
| Umax ● Imax: | 2 W | |
| Umax: | 40 kV | |
| Imax: | 100 μA | |
| Used filters/elements: | Main | 30 s at 50 kV with 40 μA |
| | | Ti, V, Cr, Mn, Fe, Co, Ni, Cu, Zn, As, Se, Rb, Sr, Hf, Ta, W, Re, Pb, Bi |
| | Low | 30 s at 20 kV with 100 μA |
| | | S, K, Ca |
| | Light | 30 s at 8 kV with 250 μA |
| | | Mg, Al, Si, P, (S), Cl, (K), (Ca), (Ti), (V), (Cr) |
| | High | 30 s at 50 kV with 40 μA |
| | | Zr, Nb, Mo, Pd, Ag, Cd, Sn, Sb, Ba |
| He-atmosphere: | No | |




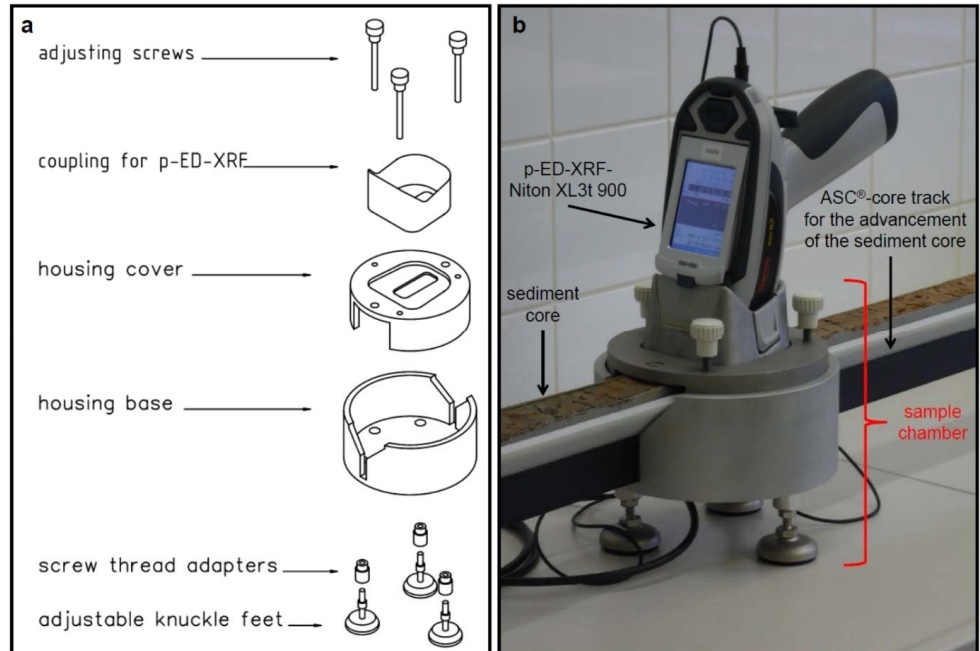

**Figure 1.** (a) Engineering drawing of the sample chamber; (b) p-ED-XRF spectrum analyser (here Thermo Scientific Niton XL3t 900) mounted to the sample chamber for measuring a sediment core within the ASC® core track.

A typical measurement cycle with the p-ED-XRF spectrum analyser starts after the core description, which is a premise for the adequate choice of certified reference material (CRM) for the calibration of the XRF-spectrometer, and consists of the calibration itself, the semi-continuous analyses of the core or sample, followed by an evaluation of the results (Fig. 3).

As CRM lake sediment LKSD-2 (Lynch 1990) was prepared into a sample cup and measured under identical measurement conditions. Only elements that show mean values larger than four times the 2 sigma error of the measurements (Al, Ca, Cl, Fe, K, Pb, Rb, S, Si, Sr, Ti, V, Zn) were taken into account for the analyses (Schwanghart et al., 2016). Typical recovery values for these elements of a lake sediment (CRM LKSD-2, Lynch 1990) and a stream sediment (CRM GBW07312) vary between 88.6 to 116.9 % depending on the element and its

15 concentration (Fig. 4). For the CRM LKSD-2 we obtained the following recovery values for the elements that were used in our study (in brackets certified concentrations):

| | | |
|---|---|---|
| SiO2 | (58.9 %) | 101.2 to 103.2 % |
| CaO | (2.2 %) | 100.5 to 102.5 % |
| Fe2O3 | (6.24 %) | 99.6 to 101.0 % |
| S | (0.14 %) | 89.3 to 109.3 % |
| Ti | (3460 µg/g dw) | 98.0 to 100.9 % |

20





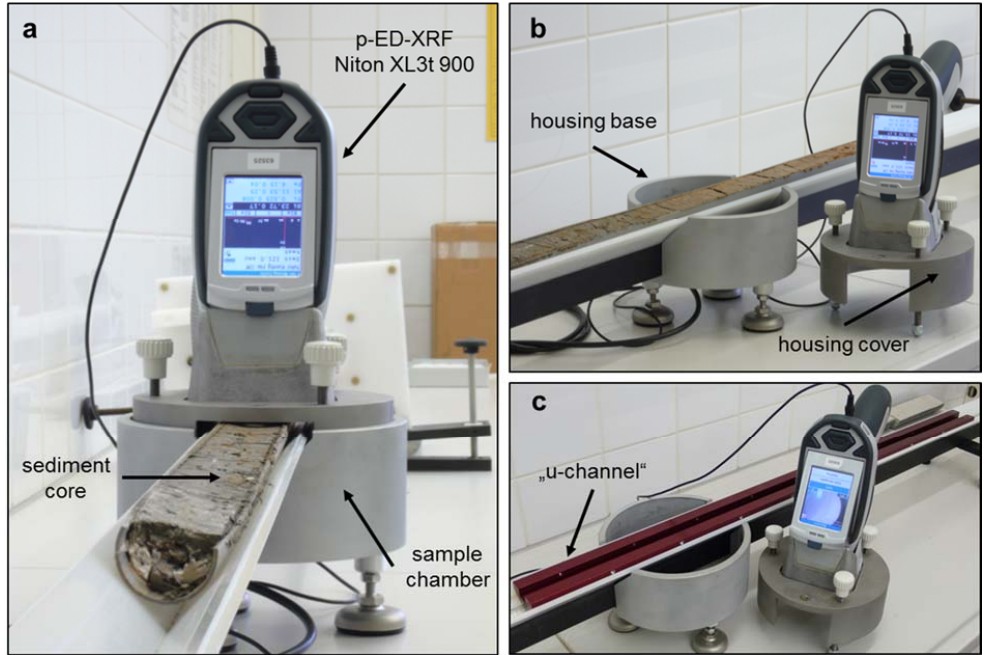

**Figure 2.** (a) p-ED-XRF spectrum analyser fixed to the sample chamber to analyze a split sediment core (a); (b) view of the housing base with the lead-through for the split sediment core and the p-ED-XRF spectrum analyser fixed to the housing cover; (c) same as (b) but with a "u-channel" often used in palaeolimnological investigations.

**Figure 3.** Schematic flowchart for a typical measurement cycle.



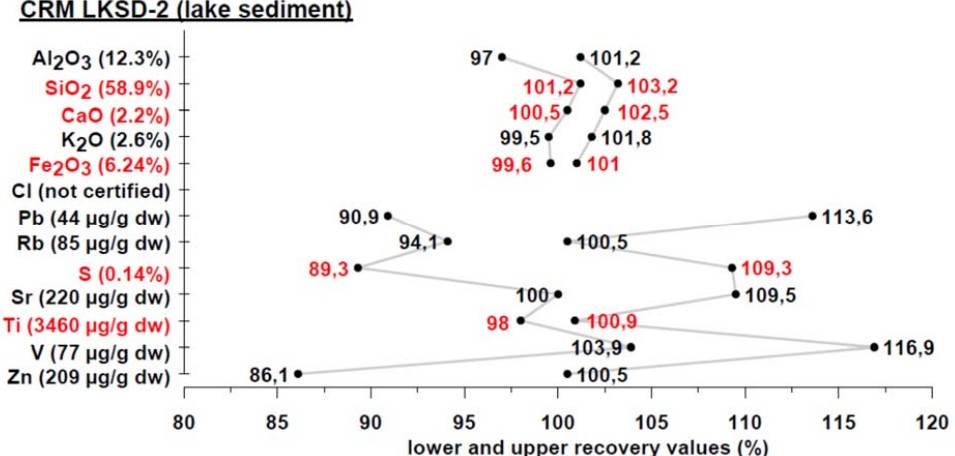

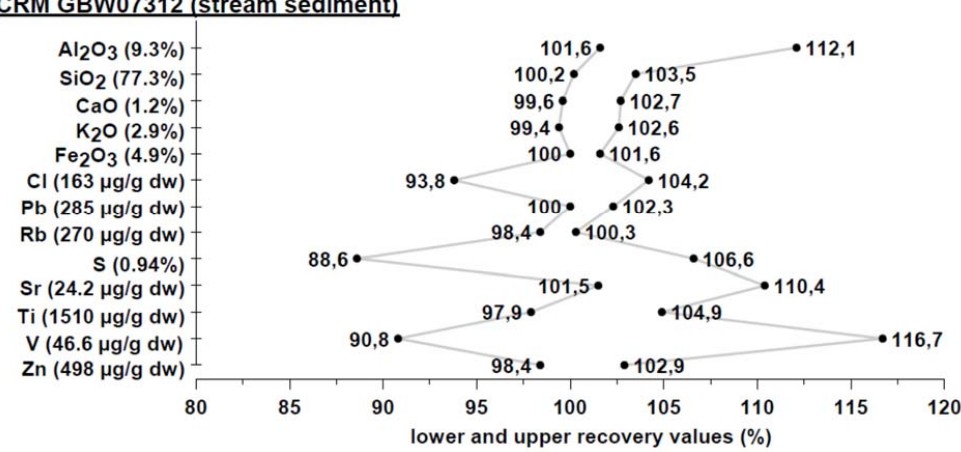

**Figure 4.** Recovery values (%) obtained for CRM LKSD-2 (lake sediment) and CRM GBW07312 (stream sediment) by p-ED-XRF analyses. The specified values are shown to the left in parentheses. The values for the elements that were used in the presented study are shown in red.

In our example, in total 823 measurements were performed with the mounted p-ED-XRF spectrometer for the 900 cm long core AYA05. To enhance accuracy each single measurement took 120 s while manual replacement of the core for the subsequent analysis is conservatively calculated at 30 s, so that the overall measurement time totals approximately 35 hrs.

**3 Results and discussion**

**3.1 Facies interpretation of core AYA05 using p-ED-XRF analysis**


Three units (unit I from 9.00m b.s. to 2.30 m b.s.; unit II from 2.30 m b.s. to 0.93 m b.s.; unit III from 0.93 m b.s. to the top of the core) were macroscopically distinguished due to sedimentary characteristics (Klein et al. 2016, in press.) complemented by elemental (TOC content) and p-ED-XRF analysis (Ca, Fe, Ti, Si, S) (Fig. 5). The latter was used as an integral measure for facies discrimination. To account for sedimentary inhomogeneity and the

varying accuracy of the element concentrations obtained by p-ED-XRF analysis we mainly focus on the interpretation of element ratios to obtain a dimensionless relative tool for the interpretation of different stratigraphic units and facies.

In coastal systems Ca input is mainly due to (1) biogenic productivity of calcium carbonate under salt-water marine conditions and as (2) a dissolved component in water. Fe enriches in terrestrial environments during weathering

and soil formation and in sediments predominately indicates hill wash processes in the drainage basin (Schütt, 2004). The Ca-Fe ratio can thus be used to differentiate geochemically between marine and terrestrial environments where high Ca-Fe ratios represent marine sediments and low Ca-Fe ratios point to terrestrial environments (Vött et al. 2011). XRF-measured Ca-Ti ratios are useful for stratigraphic correlation between cores even when the variability in concentrations is low (Hennekam and de Lange, 2012), as in core AY05. Si, Ti like

Fe reflect siliclastic terrigenous input (Arz et al., 1998, Ramirez-Herrara et al., 2012) and thus lowered Ca-Si and Ca-Ti ratios also indicate a more terrestrial origin of the sediments. The higher availability of sulphate in seawater than in freshwater results in higher S contents within brackish/marine depositional environments (Casagrande et al., 1977) and S is used as a geochemical marine marker (Chagué-Goff et al., 2012).

### 3.1.1 Unit I

Unit I extends from the base of the core to 2.30 m b.s. and consists of homogenous, weakly compacted pure silt with distinct layers of marine fauna, macro-botanical remains and charred wood. The organic carbon content with 1.4 to 2.2 mass-% (mean = 1.8 mass-%; n=67) is relatively high when compared to units II and III. Also the inorganic carbon content is generally slightly higher than in units I and II (mean = 0.28 mass-%; n = 67) and shows highest values up to 4.54 % where shell layers or limestone fragments occur. The Ca-content shows mean values

around 1.0 mass-% (std. = 0.84, n = 638) varying between 0.1 and 17.2 mass-%. The highest Ca-values are measured within the four shell layers and at a depth of 796 cm due to a limestone fragment. The slightly increased TIC- and Ca-contents correspond to marine shell fragments. The other elements measured with the p-ED-XRF spectrometer show much lower variation and mean values (n = 638) and are as follows: Fe = 2.25 %; S = 0.69 %; Si = 1.36 % Ti = 0.068 %. The ratios Ca-Fe, Ca-Ti, Ca-S, and Ca-Si show only subtle changes and point to stable

conditions in an aquiferous basin partly separated from the open sea (Dabrio et al., 2000).

The high contents of organic matter and the low contents of inorganic carbon together with the low Si contents and the grayish colour indicate oxygen-free, reducing conditions and are interpreted as a sub-tidal facies when silt as a fine-grained autochthonous lagoonal deposit was accumulated under sub-aqueous quiescent depositional conditions in a landward lagoonal-like environment.

### 35  3.1.2 Unit II

Unit II extends from 2.30 to 0.93 m b.s. and consists of moderately compacted, carbonate-free, grayish brown pure silt with hydromorphic features. The TOC (mean = 0.84 mass-%; n = 14) and TIC contents (mean = 0.08 mass-%;





**Figure 5.** Core AYA-05 with radiocarbon dates, description of the core and facies interpretation (Unit I to Unit III) based on p-ED-XRF results. a) p-ED-XRF results for Ca, Fe, Ti, and S together with TOC (total organic carbon) contents that were analyzed by elemental analysis; b) Four point running means (4prm) of Ca-Fe, Ca-Ti, Ca-S, and Ca-Si ratios. Note the logarithmic scale of the x-axis. For a detailed description of the core AYA-05 refer to Figures S1-S2 in Supplementary Data and to Klein et al. (2016, in press).



n = 14) are lower than in the underlying unit I and TOC shows decreasing values towards the top. In contrast to the carbon contents the Ti- and Si-contents increase within unit II towards the top, as is best seen in the Si-content which increases from 1.5 % at the unit's base to 5 % in the top layers of unit II. The Ca-content (mean = 0.56 %; n = 117) is generally lower than in the underlying unit I without showing any trend within unit II. Thus the Ca-Ti and Ca-Si ratios show a decreasing trend within unit II. The Ca-Fe and Ca-S ratios are lower than for unit I due to the lower Ca-content and similar Fe- and S-contents throughout unit II.

The decrease in Ca content and TIC most probably results from a reduced input of marine organisms (Engel et al., 2010) in consequence of silting-up phases as pointed out by the decreasing trends of the TOC content together with the Ca-Ti and Ca-Si ratios towards the top of unit II. The brown and silty sediments with the same grain size distribution as unit I are interpreted as salt-marsh deposits where the mottled character of this unit indicates oxidizing conditions and backs up the interpretation of reduced TOC-contents being due to microbial decomposition within an intertidal palaeoenvironment (Weston et al., 2011).

### 3.1.3 Unit III

The uppermost unit III extends from 0.93 m b.s. to the top and consists of moderately compacted, brown pure silt. Precipitation of iron oxides, charred wood and shell fragments from gastropods and molluscs are present throughout unit III. The TOC-content increases towards the top from c. 0.60 to 1.17 mass-%, whereas the TIC-contents remain low, similar to unit II. The trend of increasing Ti- and Si-contents that began in the underlying unit II accelerates within unit III and highest values are reached at the top of core AYA05. Mean Si-contents (8.34 %; n = 67) are much higher than the underlying two units and highest values reach 21.0 % Si at the top. The Ti-concentration curve parallels the Si-contents and reaches highest values near the top with 0.24 % Ti. In consequence of the higher Si and Ti contents, Ca/Si and Ca/Ti ratios further decrease in the lower part of unit III before this trend ends, probably due to proceeding soil formation and colluvial deposition. This is corroborated by the Fe-contents that remain similar to unit II except for the uppermost samples where higher iron input results from soil forming processes. Unit III documents the final stage of lagoonal infilling that is followed by soil formation and colluvial deposition and the influence of terrestrial process dynamics (Klein et al. 2016, in press).

### 3.2 Landscape development

Core AYA05 provides a continuous record of 900 cm of sediment accumulation and an assessment of Late-Holocene processes of sedimentation during the latter phases of postglacial sea-level rise. The high-resolution lithostratigraphical and geochemical study identifies three different palaeoenvironments that span the past 8000 years (Klein et al. 2016, in press). Although on generally low levels, elevated Ca-Fe, Ca-Ti, Ca-Si, and Ca-S ratios throughout unit I point to sub-aqueous depositional conditions in a landward lagoonal-like environment. After the onset of relative sea-level deceleration around 3000 cal BCE (Delgado, et al., 2012), the transition from marine (unit 1) to amphibic (unit II) conditions took place. Core AYA05 already shows an advanced stage of siltation at that stage – in contrast to other cores investigated in the study area (Klein et al., 2016 in press) – probably due to dynamical lateral changes of the tidal network. The silting-up development is geochemically represented by lowered Ca contents and an increase in Si and Ti and thus decreasing Ca/Fe, Ca/Ti, and Ca/Si ratios within unit II. This detrital input resulted presumably from human-induced deforestation and soil erosion (Fletcher et al. 2007;

Mendes et al., 2010). This development accelerates in the uppermost 93 cm of core AYA05 (unit III) and remains of charred wood become more abundant, pointing to colluvial redeposition and increased human pressure on the landscape. At this stage the maritime activities of Phoenician settlers (~1000 cal BCE) were restricted to the westernmost area directly facing the Guadiana River (Morales, 1997; Klein et al. 2016) as the central core AYA05

and the easternmost core AYA04 were already part of the amphibic zone. The continuous aggradation of organic-rich silt sediments led to a final and complete siltation towards supra-tidal conditions around 1000 CE, as documented in Klein et al., ( in press 2016).

**4 Conclusions**

The new sample chamber described here allows a portable XRF spectrometer to be used to perform quasi-

continuous elemental analyses of a sedimentary sequence of split sediment cores. In this example of use, the results of element concentrations (mainly Ca, Fe, Ti, Si, and S) and ratios (Ca-Fe, Ca-Ti, Ca-Si, Ca-S) were selected to (i) identify stratigraphic units, (ii) differentiate facies changes, and (iii) determine the Holocene sedimentary history of a marine embayment under estuarine conditions. The more than 800 XRF-analyses were executed nondestructively without elaborative sample preparation or expensive consumables directly on the dried split

surfaces of the sediment cores. The selected element concentrations were monitored and quantified for sufficient accuracy by analyzing certified reference material. This new method has been proved to serve as a strong alternative tool where cost-intensive computer-controlled core scanning XRF devices are not available or where relative high sedimentation rates require the application of basic XRF-instruments with much lower spatial resolution between 0.3 cm * 0.3 cm to 0.8 cm * 0.8 cm.

Common portable XRF devices mounted with adapters to the sample chamber introduced here (German industrial property right no. 20 2014 106 048.0) facilitate the low-cost, non-destructive, fast, and semi-continuous analysis of (sediment) cores or other solid samples.

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

**Appendix A**

The sample chamber can be adapted to mount any portable XRF-device and is available at the Institute of Physical

Geography at the Freie Universität Berlin, Malteser Strasse 74-100, 12249 Berlin, Germany.

**Acknowledgements**

The authors thank the excellence cluster EXC264 Topoi (The Formation and Transformation of Space and Knowledge in Ancient Civilizations) of the Deutsche Forschungsgemeinschaft (DFG) for providing funds for the construction of the sample chamber as well as for the fees linked to the German industrial property right no.

147GM2904. We are also grateful for the support facilitated by the Patent and Licensing Service (PULS) of the Freie Universität Berlin. The sediment cores were provided by the project A-1-7 "Ayamonte" of the EXC264 Topoi of the Deutsche Forschungsgemeinschaft (DFG).