# Peer review of "A new device to mount portable energy dispersive X-ray fluorescence spectrometers (p-ED-XRF) for semi-continuous analyses of split (sediment) cores and solid samples"

_Geoscientific Instrumentation, Methods and Data Systems, 2016_

## Referee Comment (RC1) · Anonymous Referee #1 · 17 Dec 2016

General Comments: The manuscript describes a new device for on-site analysis of split sediment cores by x-ray fluorescence with portable state-of-the art instruments to which it appears to be easily adaptable. The authors convincingly illustrate that such a simple adaptor may be very useful in many field studies of geo-archaeological investigations. This justifies publication in a journal devoted to Geoscientific Instrumentation.

Specific Comments: Prior to publication, some clarifications and changes are recommended: The present version seems to be the instrumental section of a more general publication in Quaternary International, in this text referred to as "Klein et al. (2016 in

press)". It is recommended to clearly state the relation to that article in the introduction. It may be helpful also in the sense that, in case some of the used terms and/or abbreviations are not explained, a more detailed description/explanation may be found there.

Besides the close link to the article in press "Klein et al., 2016" which is about the geoscientific research result in more general and basic terms, the manuscript presented here should be a contribution on "its own", by which it is meant that it should be inherently consistent. For that purpose a few explanations should be included which one only finds mentioned in the reference (Klein et al., 2016) and not here, like:

1. define b.s. (below surface?) or below sea level?,

2. define TOC (Total organic carbon?),

3. define TIC (Total inorganic carbon?).

Additionally:

4. Since the exact arrangement of most likely stacked foils for use as appropriate filters for different element regions of the periodic system of the elements is perhaps not revealed by the NITON company, at least a general description of the choice of filter and voltage combination to enhance certain elements or groups of elements should be added to the description of the XRF measurements. Because of the rather sophisticated filtering for enhancing different elemental groups extracting concentrations, as given e.g. in Fig.5, may not be straight forward.

5. A description of how the other elements, not detected by XRF like C, are determined, is missing here in the manuscript, but given in (Klein et al., 2016), at least a word must be said about it.

Technical corrections: 1. The authors should find a way to avoid stating the "German industrial property right No. etc." in all detail repeatedly.

2. There should be a common way to give a reference, it appears to be as (author et al.,year). It should be used consistently, not varying. If "in press" has to be added, it may be sufficient to add this comment once, in the list of references.

3. p.2, line 12: instead of "ca." use "approximately".

4. P.7, line 24: "higher than in units I and II. . ." sounds inconsistent under the subtitle 3.1.1. Unit I.

5. p.10, line 18: relatively.

Final remark: In their conclusion the authors may add, as kind of an outlook, that this simple adapter for a p-XRF may easily be computer controlled with a stepping motor (even on-site) to simplify the scanning step-by-step in 1-cm steps over 35 hours (time given in the manuscript).

---

## Author Comment (AC1) · 21 Dec 2016

Author's response: We thank the anonymous referee #1 for his valuable comments that help to improve our manuscript. We incorporated the proposed changes of referee #1 and report these herewith.

In the manuscript we now clarified the close link to the article Klein et al. (2016) and our intention to describe the technical and methodological details of the newly introduced sample chamber and to present an application example (p. 2, lines 09-12).

- define b.s. => p. 2, line 35 - define TOC => total organic carbon; p. 8, line 02 - define

[Figure]

TIC => total inorganic carbon; p. 7, line 13

A description of the other elements not detected by XRF, like C have been added to the manuscript (p. 6, line 11 – p. 7, line 04).

We have difficulties to understand comment #4 of referee #1 that asks for a general description of the choice of filter and voltage combinations to enhance certain elements or groups of elements. In our opinion this information is described in detail in the section 2.3 Methods (p. 3, line 27 to p. 4, line 04). The specification of the foils used as appropriate filters for the different elements is closed information and not made public by the NITON company. If additional information is needed, we kindly ask referee #1 to be more specific. However, in section 2.3 we made clear that stacked foils were used as filters for the measurements "low", "light", and "high" (p. 3, line 31 to p. 4, line 01). All technical corrections of referee #1 were implemented. We avoided in the new version of the manuscript to state the "German industrial property right number" in all detail repeatedly and now mention the details only in the abstract, the introduction, the conclusions, and the acknowledgements. A final remark as kind of an outlook to a possible automation has been added to the manuscript (p. 12, line 21-23).

All corrections in the new version of our manuscript are marked in the attached supplementary file.

Please also note the supplement to this comment:
http://www.geosci-instrum-method-data-syst-discuss.net/gi-2016-33/gi-2016-33-AC1-supplement.pdf

**Supplement:**

**A new device to mount portable energy dispersive X-ray fluorescence spectrometers (p-ED-XRF) for semi-continuous analyses of split (sediment) cores and solid samples**

Philipp Hoelzmann[1], Torsten Klein[1], Frank Kutz[1], Brigitta Schütt[1]

[1]Physical Geography, Institute of Geographical Sciences, Department of Earth Sciences, Freie Universität Berlin, Berlin, Germany

*Correspondence to*: Philipp Hoelzmann (philipp.hoelzmann@fu-berlin.de)

**Abstract.** Portable energy-dispersive X-ray fluorescence spectrometers (p-ED-XRF) have become increasingly popular in sedimentary laboratories to quantify the chemical composition of a range of materials such as sediments, soils, solid samples, and artefacts. Here, we introduce a low-cost, clearly arranged unit that functions as a sample chamber (German industrial property right no. 20 2014 106 048.0) for p-ED-XRF devices to facilitate economic, non-destructive, fast, and semi-continuous analysis of (sediment) cores or other solid samples. The spatial resolution of the measurements is limited to the specifications of the applied p-ED-XRF device – in our case a Thermo Scientific NITON XL3t p-ED-XRF spectrometer with a maximum spatial resolution of 0.3 cm and equipped with a charge-coupled device (CCD)-camera to document the measurement spot. We demonstrate the strength of combining p-ED-XRF analyses with this new sample chamber to identify Holocene facies changes (e.g. marine vs terrestrial sedimentary facies) using a sediment core from an estuarine environment in the context of a geoarchaeological investigation at the Mediterranean coast of southern Spain.

**1 Introduction**

Bulk-sediment chemistry data of split core sediment surfaces provided by non-destructively measured portable X-ray fluorescence is important for the determination of sedimentary composition, the interpretation of varying facies, and thus for the reconstruction of environmental conditions. Therefore, computer-controlled core scanning XRF tools are often successfully applied (Jansen et al. 1998; Koshikawa et al. 2003; Kido et al. 2006; Richter et al. 2006) and supply bulk-sedimentary data at nearly continuous µm-scale resolution, first of all for environments that provide highly resolved archives with low sediment accumulation rates such as in cores from marine or lacustrine environments. However, these instruments are extremely cost-intensive and especially terrestrial sediments often show a lower temporal resolution with much higher sediment accumulation rates in the range of centimetres per year that allow the application of basic XRF-instruments with much lower spatial resolution.

This paper introduces a hands-on, low-cost device (German industrial property right no. 20 2014 106 048.0) that uses common adapters to mount p-ED-XRF devices so that these can provide bulk-sedimentary chemistry data from non-destructive measurements at the surface of a split sediment core or from other solid samples at a spatial

resolution between 0.3 cm * 0.3 cm to 0.8 cm * 0.8 cm. Our device bridges the gap between sophisticated, computer-controlled core scanning XRF systems with highest spatial resolution and simple manually performed hand-held p-ED-XRF analyses. We demonstrate the strength of combining p-ED-XRF analyses with this new sample chamber to identify Holocene facies changes (e.g. marine vs terrestrial sedimentary facies) using a

5 sediment core from an estuarine environment in the context of a geoarchaeological investigation at the Mediterranean coast of southwestern Spain (Klein et. al. 2016).

At the archaeological site Ayamonte (S-Spain), located at the estuary where the Guadiana River flows into the Atlantic in the southwest of the Iberian Peninsula, sediment archives were investigated in order to provide input for an improved environmental reconstruction of the Holocene settling phases. In this study we focus on (i) the

10 technical description of the newly introduced device; (ii) how bulk-sedimentary chemistry data was 
[revised manuscript text omitted]

10 The sediment color was determined using a Munsell soil color chart. The TC (total carbon) content was analysed with a LECO TruSpec CHN analyser by dry combustion at 950˚C of 0.1 g of homogenised and dried sediment samples in an oxygen atmosphere followed by IR-detection of the evolved $CO_2$. For the determination of the total inorganic carbon (TIC) content $CO_2$ was evolved during hot (70˚C) acid ($H_3PO_4$) treatment from 0.1 g of dried and homogenised sediment samples. Quantification was performed with a Woesthoff Carmhograph C-16

by measuring the change in conductivity caused by the evolved $CO_2$ in 20ml of a 0.05N NaOH solution. The total organic carbon (TOC) was calculated by subtracting TIC from TC.

[revised manuscript text omitted]

---

## Referee Comment (RC2) · Anonymous Referee #2 · 23 Dec 2016

Hoelzmann and co-authors present a new device to mount a handheld XRF scanner for core logging purposes. This development has potential for low cost analyses down to cm-resolution and therefor useful for publication. However, some aspects of the current paper need some additional attention before.

The introduction opens with a statement of XRF scanning ending with ". . . lower temporal resolution with much higher sediment accumulation rates in the range of centimeters per year that allow the application of basic XRF-instruments with much lower spatial resolution." However, the wide application of XRF scanners on marine sediment

cores proves that this is incorrect. The power of (XRF) core scanning methods is that these applications provide continuous geochemical records, whereas the resolution rather depends on the type of sediments (e.g. fine laminated or homogeneous) not of the sedimentation rate. The introduction and first part of the methods (part 2.1) need some additional attention to discard doubling and improve. Also, in line 3-6 the site is described as being part of geo-archeological research at the Mediterranean coast, but it is situated at the Atlantic coast.

The figures or the methods are partly redundant or repetitive, whereas some technical aspects are not mentioned. For example there is no indication of the distance between if a foil is used to cover the core during analyses. Some specific part of the equipment presented here are "protected" by a German Industrial Property right no., which is cited four times in the text and once as Anonymous, 2014. I suggest indicating this part once (including the patent number) in one of the figures is sufficient.

The first paragraph of the discussion suggests that sediment in-homogeneity is probably causing bias on the geochemical records presented in figure 5 (in this case both precision and accuracy). Ratios of elements are suggested to reduce the bias effectively as shown in figure 5b. However, this relation is not new and has already been suggested by (e.g.) Jansen et al. (1998), Richter et al. (2006) and Rothwell et al. (2006). Moreover, Weltje & Tjallingii (2008) described as bias originating by sample geometry effects and accommodated this as part of their log-ratio calibration model. Previous work on the bias generated by sediment in-homogeneity should be acknowledged.

The authors state that S can be used as a marine indicator, which I doubt is a reliable indicator due to the partly high amount of organic matter. Marine influences in coastal sediments have been described by the elements Br and Cl as these elements are contained in seawater (e.g. Mayer et al., 2007, Boer et al., 2015)

33, 2016.

---

## Author Comment (AC2) · 9 Jan 2017

Author's response to referee#2:

We thank the anonymous referee #2 for his valuable comments that help to further improve our manuscript. We incorporated the proposed changes of referee#2 and report these herewith. In the introduction we now made clear the relation between the often lower temporal resolution of terrestrial records that may justify the application of basic XRF instruments which have a much lower spatial resolution (page 1; lines 27-29).

[Figure]

In section 2.1 (Study Site, Material, and Methods) discarded doubling and corrected for the archaeological site which is indeed situated at the Atlantic coast (page 1; line 19).

The number of figures were not reduced as proposed by referee#2 because the editor advised to add the included figures to the first submitted version of our manuscript.

Within the methods part we added information concerning the distance between the p-ED-XRF-device and the object (split sediment core) and pointed out that the measurements can be performed with or without a foil as protection (page 3; lines 15-19).

The citation of the German Industrial Property Right No. is limited now to the Abstract and caption of Fig. 1 (page 1; line 13 / page 4; line 10).

References were added to show that ratios of elements were used to effectively reduce the bias that might be due to sediment in-homogeneity: Jansen et al., 1998; Richter et al., 2006; Rothwell et al., 2006; Weltje and Tjallingii, 2008 (page 7; line 9).

We added a second reference that shows that sulfur can be used as a marine indicator (Chagué, 2010; page 7 line 20). We thank referee#2 for the suggestion to consider Br and Cl as marine indicators in coastal sediments. Unfortunately, we could not identify the references suggested by referee#2 (Mayer et al, 2007; Boer et al, 2015) without further information (additional co-authors and/or journal information). Br was not measured with the p-ED-XRF spectrometer and therefore also not mentioned in the Method's section (2.1).

We now included Cl in Figs. 4 and 5 (page 6 and page 8) as it is often used as an indicator of marine conditions (Chagé-Goff et al., 2012) and changed the captions accordingly. However, the interpretation of Cl in our semi-terrestrial to coastal sediments is difficult, probably due to the influence of groundwater circulation and the high solubility of Cl. This is shown in relatively large changes in the Cl content throughout all units without showing any clear trend (page 7; lines 20-24 and lines 36-37). It is with referee#2 to decide if the new version of Figs. 4 and 5 (with Cl) or the former version

of Figs. 4 and 5 (without CI) should be used.

All newly added references were also added to the list of references.

All corrections in the new version of our manuscript are marked in the attached supplementary file which was produced with the "track change mode".

Please also note the supplement to this comment:
http://www.geosci-instrum-method-data-syst-discuss.net/gi-2016-33/gi-2016-33-AC2-supplement.pdf

**Supplement:**

[revised manuscript text omitted]

10 The sediment color was determined using a Munsell soil color chart. The TC (total carbon) content was analysed with a LECO TruSpec CHN analyser by dry combustion at 950°C of 0.1 g of homogenised and dried sediment samples in an oxygen atmosphere followed by IR-detection of the evolved $CO_2$. For the determination of the total inorganic carbon (TIC) content $CO_2$ was evolved during hot (70°C) acid ($H_3PO_4$) treatment from 0.1 g of

dried and homogenised sediment samples. Quantification was performed with a Woesthoff Carmhograph C-16 by measuring the change in conductivity caused by the evolved $CO_2$ in 20ml of a 0.05N NaOH solution. The total organic carbon (TOC) was calculated by subtracting TIC from TC.

**3 Results and discussion**

**3.1 Facies interpretation of core AYA05 using p-ED-XRF analysis**

[revised manuscript text omitted]